# Nutrition in Patients with Inflammatory Bowel Diseases: A Narrative Review

**DOI:** 10.3390/nu14040751

**Published:** 2022-02-10

**Authors:** Leda Roncoroni, Rachele Gori, Luca Elli, Gian Eugenio Tontini, Luisa Doneda, Lorenzo Norsa, Marialaura Cuomo, Vincenza Lombardo, Alice Scricciolo, Flavio Caprioli, Andrea Costantino, Lucia Scaramella, Maurizio Vecchi

**Affiliations:** 1Center for Prevention and Diagnosis of Celiac Disease, Gastroenterology and Endoscopy Unit, Fondazione IRCCS Ca’ Granda Ospedale Maggiore Policlinico, 20122 Milan, Italy; rachele.gori@studenti.unimi.it (R.G.); luca.elli@policlinico.mi.it (L.E.); gianeugenio.tontini@policlinico.mi.it (G.E.T.); vincenza.lombardo@policlinico.mi.it (V.L.); alice.scricciolo@policlinico.mi.it (A.S.); flavio.caprioli@unimi.it (F.C.); andrea.costantino@policlinico.mi.it (A.C.); lucia.scaramella@unimi.it (L.S.); maurizio.vecchi@policlinico.mi.it (M.V.); 2Department of Biomedical, Surgical and Dental Sciences, University of Milan, 20100 Milan, Italy; luisa.doneda@unimi.it; 3Department of Pathophysiology and Transplantation, University of Milan, 20100 Milan, Italy; 4Pediatric Hepatology Gastroenterology and Transplantation, Papa Giovanni XXIII Hospital, 24127 Bergamo, Italy; lonorsa@hotmail.com; 5Department of Pediatrics, San Carlo Borromeo Hospital, ASST Santi Paolo e Carlo, 20142 Milan, Italy; mariala.cuomo@gmail.com

**Keywords:** IBD, Mediterranean diet, gluten, lactose, FODMAP diet, nutrition, pediatric patient

## Abstract

Inflammatory bowel diseases (IBD) affect the gastrointestinal tract: they include Crohn’s disease (CD) and ulcerative colitis (UC). Each has a different phenotypic spectrum, characterized by gastrointestinal and extra-intestinal manifestations. People living with IBD are very interested in diet, but little is known about the impact of diet on these patients; no guidelines are available yet. In this review, we analyze the dietary patterns of patients with IBD and the approach to the choices of foods both in adults and pediatric patients. Very often, IBD patients report an intentional avoidance of gluten to manage the disease; furthermore, a proportion of IBD patients believe that dairy products worsen their symptoms and that avoidance may help the disease. They have a low compliance with the Mediterranean Diet, which is considered to have potential benefits but is little used in practice. In conclusion, the review underscores the pivotal role of nutritional counselling in IBD patients, and the importance of future clinical studies to evaluate the beneficial effects of dietary recommendations in the management of IBD.

## 1. Introduction

Inflammatory bowel diseases (IBD) are a heterogeneous group of inflammatory disorders of the gastrointestinal (GI) tract, with a chronic or recurrent clinical course, characterized by phases of exacerbation and remission. The clinical presentation has a heterogeneous phenotypic spectrum, characterized by gastrointestinal and extraintestinal manifestations, with atypical or non-specific symptoms. Crohn’s disease (CD) and ulcerative colitis (UC) are the most typical conditions in this spectrum, and they differ in anatomical location and type of lesions. The etiology is multifactorial [1], and to date, the prevailing hypothesis is that an abnormal mucosal immunological response is triggered against ubiquitous antigens, such as the resident bacterial flora, in genetically predisposed individuals. However, since the 1970s, in the pathogenesis of IBD diseases [2], there has been an increase in the number of diagnoses, especially in industrialized countries [3], thus highlighting the potential role of environmental factors, such as lifestyle and dietary habits, the interaction between diet and the susceptibility to gene variants, abnormal gut microbiota and altered immune response [1].

IBD significantly affects the quality of life of patients and their families, as lifelong modification of behavior, lifestyle and eating habits is required [4]. IBD management often consists of the use of immunomodulators and immunosuppressive drugs to control active flares and maintain a state of remission [5]. Dietary changes might be helpful in reducing symptoms, such as abdominal pain or diarrhea, and lessening the effects of IBD complications [6]; however, to date, the impact of diet on the disease is still highly debated in literature, and there are no dietary guidelines. This lack of clear information pushes patients to seek information from other sources and ultimately increases the risk of self-imposed dietary restrictions with important negative consequences on patients’ health [7]. The dietary habits and intake of patients with IBD can also be based on personal experiences that may lead to the avoidance of certain foods or food groups to self-manage the disease. In this review, we analyze the dietary patterns of patients with IBD and how these patterns change according to the severity of the disease.

## 2. Role of Diet in the Pathogenesis of IBD

Diet has an important impact on microbial composition, the morpho-functional integrity of the intestinal barrier and host immunity. Changing the intake of specific food groups may promote gut dysbiosis; this leads to an alteration of the gut barrier, immune activation and tissue damage, and can have a role in the development of IBD [8]. The Western diet, which is high in saturated fat, red meat and refined sugars, and low in fiber, fresh fruit and vegetables is considered a possible risk factor in the development of IBD [7]. In any complex disease, where diet is a contributing factor, it is difficult to understand the role of any single food, because dietary patterns involve exposure to different groups of foods. To date, evidence on how diet influences IBD activity is insufficient. Furthermore, the impact that food choices may have on the disease course remains unknown, with the exception of exclusive or partial enteral nutrition, which has been shown to induce remission in patients affected by CD [6,9]. Table 1 provides a summary of epidemiological studies which have investigated the function of food groups in the development and evolution of IBD. A prospective study analyzed a cohort of women living in France and concluded that high animal protein consumption was significantly related to an increased risk of developing IBD [10]. Few studies have tested the connection between long-term intake of dietary fiber and the risk of development of IBD.

Intake of dietary fibers, specifically from fruit, is associated with a lower risk of CD (40% reduction in risk) [11]. Smoking, antibiotic use and diet are potentially reversible risk factors for IBD. A diet rich in fruits and vegetables, rich in n-3 fatty acids and low in n-6 fatty acids is associated with a decreased risk of developing CD or UC [12,13].

A high intake of dietary long-chain n-3 polyunsaturated fatty acids was associated with a reduced risk of UC, while high dietary intake of n-6 polyunsaturated fatty acids and trans-unsaturated fats was associated with the highest risk of UC [8,14,15]. Sakamoto et al. [16] demonstrated an adverse effect of sweets and artificial sweeteners on the risk of developing both UC and CD. However, a large prospective study has demonstrated no link between the total intake of carbohydrates and incidence of UC or CD [17]. There are conflicting studies in the literature concerning the role of carbohydrates in the progression of IBD; thus, further studies in this area are required. According to the European Prospective cohort study, no relation was found between alcohol consumption and the risk of CD and UC [18,19]. There are some epidemiological studies that demonstrate an increased risk of IBD in populations who consume high levels of red meat and saturated fat, but low levels of fiber [19,20]; for this reason, it is a general recommendation that patients with IBD limit the long-term consumption of foods that are typical of the Western diet. Various dietary approaches are available in the literature for the treatment of IBD that have shown success in controlling symptoms. These have been described in Table 1, but accurate randomized controlled trials are needed to verify their efficacy and recommend them to patients with IBD [21,22].

## 3. From Malnutrition to Psychological Disorders in Patients with IBD

Despite the importance of diet in IBD, today the data available to healthcare professionals to provide evidence-based recommendations are limited. This leads patients with IBD to seek support from other resources, especially from the Internet and the media. These patients’ nutritional choices are also based on personal experiences; they avoid certain foods that might exacerbate their symptoms. However, the self-management of elimination diets implies an increased risk of malnutrition and of restrictive eating disorders. A high degree of psychological disorders (depression, 91%) and malnutrition (59%) were found in IBD patients [33]. Indeed, IBD patients are characterized by an inflammatory state that induces structural and functional changes in the bowel, which can lead to reduced absorption in the gut and thus to weight loss and micronutrient deficiencies [5,7,21]. Dietary choices of patients with IBD may contribute to the state of malnutrition, due to self-imposed limits on certain foods in order to manage their symptoms. The prevalence of malnutrition can reach up to 85% in patients with IBD and includes protein-calorie malnutrition, micronutrient deficiency, or both [34]. The most common micronutrient deficiencies in patients with IBD are iron, cobalamin, folic acid, vitamin A, vitamin D, vitamin K, selenium, zinc, and vitamin B1 [34]. In addition to its biological role, food consumption is a cultural and social activity, and is a source of pleasure and conviviality with others. These pivotal psychosocial roles may be altered in people with IBD [35]. Eating habits can influence social life and, in some cases, patients with IBD avoid dining outside the home or do not share the same menu or diet with other household members [35]. Moreover, malnutrition is common in anorexia nervosa (AN) and especially in Crohn’s disease patients, also affecting patients with active IBD (up to 70%) and patients in remission (up to 38%) [36].

## 4. Overweight in Patients with IBD

Studies have reported a high prevalence of obesity in IBD patients; about 15–40% of patients with IBD are obese [37], contrary to conventional beliefs that patients with IBD are malnourished. In IBD patients, the high prevalence of obesity suggests a possible role of IBD in obesity development risk. However, there are not enough studies available that confirm this correlation. Obesity has been shown to be an independent predictor of low response to medical therapies of autoimmune diseases. Indeed, a risk factor associated with increased drug clearance, short half-life and low trough drug concentrations is related to high body weight [37,38]. Studies on obesity and treatment response in IBD are limited. This evidence highlights the importance of obesity in IBD patients and the need to consider treating obesity with nutritional, lifestyle, pharmacological or surgical interventions as adjunctive therapy for these patients.

## 5. Dietary Approaches in Adult IBD Patients

According to the ESPEN guidelines, there is no diet that can be generally recommended to promote remission in IBD patients with active disease [12]. A possible risk encountered in these patients is the development of nutritional deficiencies, especially in micronutrients; for this reason, IBD patients should be checked frequently. In patients with severe and frequent diarrhea, body fluids and electrolytes should be monitored. When food intake is not indicated or insufficient, Oral Nutrition Supplements (ONS) are used to help prevent dehydration and replace electrolytes. Enteral Nutrition (EN) should be considered as supportive therapy if oral feeding is not sufficient, while EN is always preferred to Parental Nutrition (PN). PN is indicated when there is an obstructed bowel and is required in patients with short bowels. In adults, EN is less useful than corticosteroids in influencing remission of active CD [39]. However, in children, EN is the first-line treatment in the induction of remission. Corticosteroids are associated with severe side effects, and it is also important to meet the child’s nutritional requirements for proper growth and health. During the remission phases of IBD, there is no specific diet. A meta-analysis has shown that probiotic treatment is effective in the active UC and remission phases, but no beneficial effect was found in the maintenance of remission in CD patients [40]. CD, mesenteric infarction, actinic enteritis, neoplasms, volvulus, and congenital anomalies are among the main causes of extensive resection of the small bowel, with or without a portion of the colon. As the jejunum is the main site of digestion and absorption of most nutrients, a surgical resection leads to the loss of an area of absorption, which significantly reduces its function. These surgical procedures result in short-bowel syndrome, a disorder characterized by an insufficient absorptive surface area in the intestine. This intestinal loss results in the malabsorption of fluids, electrolytes and other essential nutrients, severe diarrhea, dehydration, and progressive malnutrition. ESPEN guidelines recommend that short bowel syndrome (SBS) patients consume regular whole-food diets. SBS patients with a maintained colon should have a high intake of complex carbohydrates (60%) and a low intake of fat (20%) in order to increase overall absolute energy absorption [41,42]. Compared to long-chain triglycerides, a high content of medium-chain triglycerides is preferred in SBS patients with a preserved colon. In these patients, attention is paid to the potential deficiencies in essential fatty acids and fat-soluble vitamins [41]. However, the degree of evidence for these recommendations is low. It is always important that dietary counselling be conducted by a nutritionist, considering the patient’s experience, in order to meet the body’s energy and metabolic requirements and to prevent nutritional deficiencies.

### 5.1. Gluten Free Diet

Gluten is a peptide compound typically found in some cereals, especially wheat, rye, barley, and often in oats. Gluten is made up of two protein classes: glutenins and prolaminins, called gliadins in wheat and otherwise known as hordeins in barley, secalins in rye, and avenins in oats. In contrast to coeliac disease (CeD), there is no recommendation to remove gluten from the diet in IBD. IBD patients often report an intentional avoidance of gluten to manage IBD symptoms [21,43] and most of the IBD patients who decide to avoid gluten do not have a diagnosis of CeD. Some studies have investigated how the GFD may improve symptoms and/or influence the course of IBD [24,44,45]. In a large study, patients with reported IBD had to fill in a baseline survey and, if on GFD treatment, they also completed a survey on adherence to GFD. Nearly two thirds of patients who had tried a GFD reported improvement in their GI symptoms, 38.3% of patients reported less frequent or less severe IBD symptoms while on GFD, and 23.6% reported that they required fewer medications to control the disease. The authors confirmed that a trial GFD in IBD patients with significant gastrointestinal symptoms may be a safe treatment option, but advised that further prospective studies are needed to understand the mechanism of gluten sensitivity in IBD [24]. In a Swiss IBD cohort, no differences in terms of disease activity, hospitalization and surgery were found in patients consuming or not consuming gluten [25]. Therefore, there is not yet enough evidence to recommend a GFD for IBD patients. Many studies suggest a pathophysiological correlation between CeD, microscopic colitis (MC) and IBD [46,47]. Patients with concurrent IBD and CeD require lifelong compliance to GFD to prevent exacerbation of both diseases. In IBD patients, regardless of non-celiac gluten sensitivity (NCGS), gluten is a dietary factor that may influence symptoms. Long-term adherence to a GFD appears to have an impact on the psychological aspect. Conflicting results have emerged regarding the psychological state of IBD patients on a GFD; some studies have shown that a GFD was responsible for worse overall psychological well-being, while in other studies compliance with a GFD has been linked to lower rates of depression [24,25]. A study by P. Schreiner et al., was not able to observe an association between GFD and the course of IBD; a proportion of patients with IBD avoided gluten according to underlying beliefs of positive effects on disease course and not due to an objective improvement [48].

Theoretically, Gliadin activates zonulin signaling, which leads to an increase in intestinal permeability to macromolecules, contributing to the activation of the innate immune response [49,50]. Therefore, gluten could have a direct link with inflammation and IBD symptoms, although there are not enough studies at this time to support this hypothesis.

### 5.2. FODMAP Diet

Symptom management can also be linked to FODMAP reduction with a GFD. FODMAPs is an acronym, derived from ‘Fermentable, Oligo-, Di-, Mono- saccharides And Polyols’, to indicate a specific group of foods. FODMAPs are poorly absorbed in the small intestine and are subject to bacterial fermentation; for this reason, they are related to the decline or stimulation of functional gastrointestinal symptoms, such as bloating, cramping and diarrhea. It is known that a GFD usually leads to a significant decrease in dietary FODMAPs [51]. Major dietary FODMAPs are also present in gluten-containing foods, so there would appear to be a close association between a low-FODMAP diet and a GFD. However, future studies are required to further investigate the function of gluten and FODMAPs within the IBD population. Despite being naturally present in foods, a reduction in FODMAPs may be helpful in managing GI symptoms in some patients [52]. FODMAPs include also fructose and fructans; fructans are fermentable, leading to gas production in the colon, while fructose increases the volume of small-intestinal water. A case-control study showed that patients with active CD consumed inferior quantities of oligofructose and fructans compared to their inactive CD counterparts and healthy controls [53]. There are both uncontrolled and controlled studies in the literature which show a positive influence of a low-FODMAP diet in patients with IBD [54,55]. Prince et al. noticed a relevant decrease in the severity score of most symptoms (*p* < 0.001) as well as a significant improvement in frequency and stool consistency [54]. A pilot study of patients with stable IBD highlighted that a dietary intervention focusing on the reduction of FODMAPs for 3 months led to a long-lasting improvement in symptoms in patients with IBD, such as diarrhea, bloating, abdominal pain and flatulence [56,57]. The low-FODMAP diet may have a considerable beneficial impact on patients with IBD, through a decrease in fermentation by intestinal bacteria and osmotic effects.

### 5.3. Lactose-Free Diet

Dairy products (milk, yoghurt, cheese) are particularly nutritious foods, and this is demonstrated by their important position in the Mediterranean diet pyramid. They contain all of the most nutritious substances (carbohydrates, fats, proteins), the necessary vitamins (A, B, D), and important ions (calcium) needed for the structure of a healthy body. Despite their importance, they can cause intestinal symptoms, such as diarrhea, flatulence and bloating, in people who are intolerant to lactose. Lactose intolerance is a food intolerance which occurs when the activity of lactase is diminished in the brush border of the small bowel mucosa. When lactose is not digested, it can be fermented by gut microbiota, leading to intestinal symptoms. Dairy products are a commonly mentioned and avoided food group in some studies involving patients with IBD [6,21,43,58,59]. The ingestion of foods that are high in saturated fats derived from milk induces intestinal dysbiosis, with a relative abundance of a sulphite-reducing species, Bilophila wadsworthia, which has been associated with a pro-inflammatory T helper type 1 immune response and an increased incidence of colitis in mice [60]. Numerous studies have shown that a Western diet, which includes high amounts of fat derived from milk and dairy products, may be a major driver of the increasing incidence of IBD [8,14,15]. A proportion of IBD patients believe that dairy products worsen symptoms, and that dairy avoidance may help the disease [5,43]. There are some studies on the anti- and pro-inflammatory effects of milk and dairy products, but further studies need to be developed with enhanced designs and better reporting to better understand the role of these products in inflammation. An association between the consumption of lactose and IBD has not been shown [61]. The results of a recent study showed that lactose sensitivity occurs in a high percentage (about 70%) of patients with IBD. The patients underwent measurement of breath hydrogen and methane, genetic testing and their symptoms were recorded. A high lactose sensitivity in IBD could not be ascribed to bowel inflammation, since the patients were in remission [62]. Therefore, dairy intake was described as worsening symptoms. However, it may not aggravate the disease process; it may simply lead to the same symptoms that individuals without IBD would also have [43]. Avoidance of this food group exposes IBD patients to an increased risk of calcium and vitamin D deficiency, bone demineralization and malnutrition. Therefore, it is of great importance and interest to investigate the dietary choices of patients with IBD related to dairy and to conduct further studies on the symptoms related to the consumption of dairy products.

### 5.4. The Mediterranean Diet

Ancel Keys in the 1960s coined the term “Mediterranean diet” (MeDiet), following the results of an epidemiological study, known as the Seven Countries Study, which showed that the populations bordering the Mediterranean Sea, such as Italy and Greece, had a lower mortality rate and incidence of cardiovascular disease and cancer than other populations. Today, it is suggested that the MeDiet be followed as a model dietary pattern to maintain health and reduce the development of chronic diseases. Several prospective population-based epidemiological studies have shown that compliance with the MeDiet may have a protective effect against cardiovascular disease, stroke, obesity, diabetes, hypertension, several types of cancer, allergic diseases and, more recently, Parkinson’s and Alzheimer’s disease [63]. One European study showed that adherence to the MeDiet was associated with a 23% lower rate of all causes of mortality [63]. Adherence to the MeDiet was defined through scores estimating the correspondence of the dietary pattern of the study population with the traditional MeDiet. The main dietary components considered were vegetables, fruit, pulses, cereals, fish and a moderate intake of red wine with meals. The traditional MeDiet is characterized by high consumption of fruits, vegetables, cereals, pulses, unsaturated fats such as extra-virgin olive oil, nuts, a medium intake of dairy products and fish, a moderate intake of red wine, and a low intake of red meat, saturated fat and sweets. This dietary pattern is rich in antioxidant vitamins (vitamin C, vitamin E, β-carotene), minerals, natural folate and phytochemicals (flavonoids) and also appears to show an important role in reducing inflammation. The fiber-rich MeDiet promotes beneficial composition of the microbiome and metabolome, particularly with a high abundance of fiber-degrading bacteria and the production of short-chain fatty acids, which are hypothesized to play a preventive role in IBD. The randomized controlled trial, Predimed, showed that the MeDiet appears to be associated with the methylation of inflammation-related genes and exerted high regulatory effects [64]. Further studies have revealed the potential role of the MeDiet in modulating gene expression, reducing inflammatory markers and normalizing the microbiota in patients with IBD [65]. Adherence to a short-term MeDiet showed an improvement of anthropometric variables related to the development of metabolic syndrome, a reduction in liver steatosis and an improvement in inflammation and disease activity indexes in both CD and UC patients [66]. A study by Papada et al. showed a link between the MeDiet score and improved quality of life in CD patients; in particular, the MeDiet score was negatively associated with the Harvey-Bradshaw Index (HBI) and C-reactive protein (CRP), while it was positively correlated with quality of life [67]. Fecal calprotectin is an important marker which, when increased, indicates intestinal inflammation. In UC patients following a pouch surgery, high adherence to the MeDiet was associated with decreasing calprotectin levels and with decreasing risk of the development of pouchitis overtime; furthermore, the MeDiet may help reduce intestinal inflammation [68]. However, it has emerged that patients with IBD have a low adherence to the MeDiet [69], and considering the potential benefits of this dietary pattern, it would be appropriate to provide more nutritional information to patients and guide them in this direction. The MeDiet can be considered an additional approach to pharmacological therapy in reducing gut inflammation and preventing IBD [65,66]. In addition to diet, it is important to maintain a healthy lifestyle; indeed, data from three prospective cohorts have highlighted that adherence to five healthy lifestyle factors, such as physical activity, non-smoking, the MeDiet, low alcohol consumption and maintaining a normal BMI, was associated with lower mortality in older IBD patients [70].

## 6. Dietary Approach in Pediatric IBD Patients

Pediatric IBD is commonly associated with malnutrition and nutrient deficiencies [71].

Historically children and adolescents with IBD appear to have growth failure at disease presentation, especially in CD, and an association with being underweight and malnutrition has also been described. Roughly one third of newly diagnosed UC pediatric patients and half of newly diagnosed CD pediatric patients present with malnutrition [72]. There is likely multifactorial etiology that contributes to sub-optimal caloric intake including food avoidance, abdominal pain and malabsorption. Furthermore, it should be considered that a chronic inflammatory condition may cause a reduction in appetite via catabolic effects and hypothalamic weight regulation. This is particularly important since malnutrition is strictly related to poor prognosis in patients with IBD [73]. Thus, ongoing scientific literature supports the main role of nutrition and diet in children with IBD. According to a recent position paper published by the European Society of Gastroenterology, Hepatology and Nutrition (ESPGHAN) [74], a global nutritional assessment in children with IBD is a central step in the management of IBD patients, with the aim of tailoring nutritional and dietary interventions. Indeed, children with IBD should be checked for qualitative and quantitative nutrient intake on a regular basis, through a weekly record. Starting from the past decade, the relationship between diet and IBD pathogenesis has been clearly supported by scientific evidence. Besides this, several nutritional strategies have evolved over the years and are currently considered therapeutic tools, with varying degrees of efficacy and support. Concerning pediatric IBD, multiple studies and meta-analyses have proven that EEN is as effective as steroid therapy in inducing remission in children with active CD [75].

In particular, EEN may induce remission in approximately 75–85% of children with mild-to-moderate CD [76], along with superior mucosal healing, a significant decrease in inflammatory biomarkers [77] and positive benefits to growth and overall nutritional status [78]. It is important to note that the efficacy of EEN in the induction of clinical remission in children with CD was further confirmed also when compared to biological therapy [79]. This nutritional therapy is based on the use of a complete liquid formula, administered either orally or via a feeding tube, which is given as the unique source of daily nutritional requirements for 6 to 8 weeks, whilst avoiding the intake of usual solid foods [80]. Due to its excellent safety profile and its equipotential to corticosteroids in inducing remission, the European Crohn’s and Colitis Organization (ECCO) and the ESPGHAN recommend EEN as the first-line therapy for mild-to-moderate pediatric CD to induce remission, both in the first flare-up and during relapses of symptoms [81]. On the contrary, until today, there has been no evidence to support EEN as an effective therapy for active UC [26]. EEN is poorly tolerated, limiting its acceptability and clinical use [80]. Therefore, alternative and better-tolerated nutritional therapy strategies have been developed to overcome non-adherence. Aimed at pinpointing a more effective and longer-term dietary therapy, the next step has been to focus on the identification of potentially proinflammatory dietary components that may negatively affect the microbiome. From this perspective, an innovative approach, called the CD exclusion diet (CDED), was described for the first time in a case series of adults and children with CD by Sigall-Boneh et al. [82]. Patients following the CDED whole-food diet coupled with partial enteral nutrition (PEN) reached high rates of clinical response and remission [82]. The efficacy of CDED + PEN compared to EEN in inducing clinical remission in children with mild-to-moderate CD was recently shown in a multinational trial [83]. CDED was developed to exclude all industrially made packaged and processed foods that are rich in preservatives and emulsifiers, sugary drinks, dairy products, red meat and all processed meats, cereals containing gluten and gluten-free industrial products, ice cream and packaged desserts. This nutritional regimen consists of different phases with incremental varieties of food allowed and provides a consistent amount of high-quality protein and sugars. A somewhat different approach was recently developed by Svolos et al. [84], with an individualized food-based therapy (CD-TREAT). The latter consists in an ordinary solid food diet that aims to recreate as closely as possible the composition of EEN, excluding certain dietary components (i.e., gluten, lactose) and the combination of others (macronutrients, vitamins, minerals and fiber). The authors anticipated that CD-TREAT aims to mimic EEN’s effects on the gut microbiome, metabolome, inflammation, and clinical outcomes. After a course of 8 weeks of CD-TREAT with five children with active CD in a pilot study, four of the five children responded to CD-TREAT, showing efficacy in inducing clinical remission [84]. Less prominent approaches include the low-FODMAP diet and the specific carbohydrate diet (SCD). However, at present, no consensus has been reached on their use in the pediatric population due to a lack of evidence. The development of novel dietary treatments has enabled a turnaround in the treatment of pediatric IBD, especially CD. Although promising, the data need to be confirmed with future well-designed studies in order to unravel the full potential of nutritional and dietary therapies for IBD in the pediatric population.

## 7. IBD Patients’ Approach to Diet

Diet can play a key role in the etiology and symptoms of IBD. Despite this, it has been observed that patients with IBD can be divided into two groups: some patients believe that intentional avoidance of certain foods can manage symptoms and improve the disease; others do not consider dietary patterns to be an additional treatment and observe improvement only with drug treatment [15]. Qualitative semi-structured interviews with patients of IBD about the psychosocial impact of food and perceptions, eating and drinking showed that some participants made a direct link between the presence or severity of their symptoms and the type of food. These patients are constantly trying to change their diet according to how they feel after eating a certain food, while others make no changes to their diet after an IBD diagnosis [35]. Whereas only few participants used diet as their first treatment for IBD, the majority believed there should be a combination of diet and drugs. Thus, the choice of dietary patterns of patients with IBD is not a static process but evolves with the disease and symptoms. Patients’ dietary patterns are connected to their understanding of how food items influence their disease course. Dietary restrictions in many patients occur during flares and are stopped in remission. Some patients claim that they can only eat chicken and rice or liquid nutrition during a flare [5,21] Several studies have been conducted to analyze the foods that are avoided by patients with IBD, summarized in Table 2, and the list of “bad” and “good” foods differed between patients. The bad foods are considered to be triggers that can exacerbate their IBD symptoms, such as diarrhea, pain, nausea and bloating. Bad foods that are typically avoided include high-fat and spicy foods, meat alternatives, fruit, vegetables, dairy products and milk, alcohol, coffee and fizzy or carbonated drinks [4,5,21,43,69]. The dietary patterns of patients with IBD are analyzed via a dietary questionnaire and interviews, both of which provide insight into intentional food avoidance.

## 8. Summary and Conclusions

Diet can play a key role in the etiology and symptoms of IBD. Experts’ opinions support ‘healthy diets’, including the consumption of plant-based instead of animal-derived foods and whole-foods instead of refined-foods. Some of the more popular diets for IBD management include the GFD, the low-FODMAP diet, the lactose-free diet and the MeDiet. In the case of active IBD, SCD and EEN represent the most relevant choices.

International guidelines recommend no complete dietary protocol for patients with IBD, so dietary recommendations can differ among physicians and nutritionists. The majority of patients with IBD intentionally avoid at least one food or food items to improve and self-manage their GI symptoms [35,88]. The most avoided foods were spicy foods, dairy products, alcohol, fruits and vegetables and carbonated beverages; these foods were also avoided during remission and to prevent relapse. In some cases, especially in the flares period, the dietary restrictions of these patients can be very severe and can lead to a compromised nutritional state. Some patients tend to eliminate gluten spontaneously, despite not having a definite diagnosis of CeD, because they believe that gluten can exacerbate gastrointestinal symptoms [43]. Studies in the literature have shown an improvement in symptoms following a GFD; however, it is not entirely clear whether this improvement is due to the exclusion of gluten from the diet or the reduction in FODMAPs. Despite an improvement in symptoms, there are no significant differences between GFD and non-GFD patients when considering hospitalization, complications and surgery. Therefore, to date, the GFD shows a potential positive role in treating the symptoms of IBD patients, but further studies examining the role of gluten in the disease course are needed to recommend GFD as a nutritional treatment [25]. Nutrition control is needed to ensure the right intake of energy, micronutrients, especially folate, iron, calcium, vitamin B12, vitamin D and fibers, hereby guiding the diet for therapeutic benefit whilst maintaining adequate nutrition [6]. Thus, diets should be delivered by a nutritionist. A case-control study investigating nutritional status in patients with CD showed significantly lower consumption of vitamins E, K, C and calcium due to the avoidance of vegetables and dairy products. Indeed, it is important for healthcare professionals to include dietary recommendations in the routine visit to avoid self-imposed restriction diets in these patients, as this can lead to adverse effects. The potential role of dietary habits in initiating IBD or in disease relapse should be interpreted with caution; indeed, there are no studies in the literature showing a cause-effect relationship. In conclusion, there is a need for nutritional counselling in IBD patients and future clinical studies should be done to investigate the beneficial effects of dietary recommendations in the management of IBD.

## Figures and Tables

**Table 1 nutrients-14-00751-t001:** Popular diets that have been the subject of scientific studies.

Diet	Components of Diet	Recommendations
In cases of active disease		
Enteral Nutrition (EN) diet	Liquid oral nutrition supplements	EN is recommended when oral nutrition supplements are not sufficient.
		Exclusive EN (EEN) is recommended to induce remission in children and adolescents with severe active Crohn’s disease (CD) [12].
Parenteral Nutrition (PN) diet	Liquid nutrition supplements	PN is indicated in inflammatory bowel diseases (IBD) when the gastrointestinal (GI) tract is dysfunctional or in CD patients with short bowel, when there is an obstructed bowel, or when other complications occur [12].
CD exclusion diet (CDED)	Only whole foods that have been minimally processed, free from additives or other artificial substances; exclusion of some foods.	CDED has shown promise in inducing disease remission; further studies are required before recommending this diet [23].
In cases of remission disease		
Specific carbohydrate diet (SCD)	Removal of grains including wheat, barley, corn, and rice; removal of added sugar, honey and most milk products. Fully fermented yogurts allowed.	There have been several studies showing the potential benefits of this diet in improving the course of disease; however, there are not enough studies yet to recommend this diet [23].
Gluten-Free Diet (GFD)	Elimination: gluten, i.e., cereals such as wheat, barley, oats, rye.	GFD appears to lead to symptomatic improvement in these patients. To date, there is not enough evidence to recommend a GFD in IBD patients [24,25].
	Allowed: gluten-free cereals (rice, maize, buckwheat, millet), meat, fish, fruit and vegetables, legumes and dairy products.	
Low-FODMAP diet (LFD)	Elimination: short-chain carbohydrates (oligosaccharides), disaccharides, monosaccharides and related alcohols.	A low-FODMAP diet may be worth trying in patients with IBD who have ‘Irritable bowel syndrome (IBS)-type’ symptoms [26].
Autoimmune diet	Modified paleolithic diet, which excludes gluten and dairy.	Preliminary efficacy in patients with active IBD, promising but lacking significant studies [27].
Vitamin C and E supplementation		Has effects on biomarkers of oxidative stress, but this supplementation has not been shown, thus far, to have significant clinical efficacy. Currently, it is not recommended [26,28].
Vitamin D supplementation		Vitamin D deficiency may affect the cause and progression of IBD, particularly CD; low-dose vitamin D supplementation seems reasonable in all patients with CD [26,29].
Omega-3 supplementation	3–4 g daily	Currently not recommended [26,30,31].
Curcumin supplementation	2–3 g daily	Curcumin shows promise as a dietary supplement as adjunctive therapy for ulcerative colitis (UC) maintenance, but data are inconclusive; currently not recommended [26,32].

**Table 2 nutrients-14-00751-t002:** Main food items that are avoided in the IBD population according to some studies.

Food Items	Avoidance
Alcohol, salads and raw vegetables, and deep-fried foods	These foods are the most commonly eliminated during the acute phase [84,85]. Alcohol was avoided to prevent flares [21,83].
Capsaicin (spicy foods)	Up to 84.8% of the IBD population chooses to avoid spicy food to prevent disease relapse [5,43,84].
Fresh fruit, vegetables and fibres	Patients are likely to avoid fibers because they are worried about disease complications [4,5,43,69].
Milk and dairy products	Dairy products are described to be commonly excluded by patients and are the food group that is most typically avoided following a health professional’s advice [5,21,43,84,85,86].
Sweets	Sweets were associated with IBD symptom aggravation more in UC patients than in CD patients.
Meat alternatives (legumes, nuts, seeds and peanut butter)	These are the most eliminated food items in the sample of IBD patients [5,84].
Coffee and frizzy drinks	Studies report that patients exclude coffee due to the worsening of symptoms; lower coffee consumption was also reported in UC patients and patients with active disease [42,85,87].

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
