# Peer review of "Nutrition in Patients with Inflammatory Bowel Diseases: A Narrative Review"

_nutrients, 2022, doi:10.3390/nu14040751_

Round 1

Reviewer 1 Report

Abstract Line 27 - 'few are known' - rephrase. Do you mean 'little is known'?

Line 47 - rectocolitis (UC) -   please change to Ulcerative Colitis

Line 68 '2. Diet and IBD' consider rephrasing to Role of diet in the pathogenesis of IBD

line 106- 'This section may be divided by subheadings. It should provide a concise and precise 106 description of the experimental results, their interpretation, as well as the experimental 107 conclusions that can be drawn. ' Either add to this by saying the experimental results are lacking etc or remove. Doesn't flow

Line 120- '

Malnutrition is present in up to 85% of patients with IBD, and includes protein-calorie 120 malnutrition, micronutrients deficiency, or both. ' Needs reference

Line 125- 

Eating habits can 125 influence social life and, in some cases, patients with IBD decline dining outside the home 126 or do not share the same menu or diet with other household members.  - needs referencing

line 127- 

Since 1980, there 127 has been a continuous increase in the incidence of overweight and obesity

consider making this into a seperate paragraph

This section on obesity rambles a little. Please consider rephrasing and cutting down

Infliximab is weight based which may explain that no differnce in response. please references papers looking at response to adalimumab and other biologics and obesity

Line 154

'Smoking, antibiotic use and diet are potentially reversible risk factors for IBD. A diet 154 rich in fruit and vegetables, rich in n-3 fatty acids and low in n-6 fatty acids, is associated 155 with a decreased risk of developing CD or UC [20]. '

This line would be better in section 2

Line 158 I

'Despite this, IBD patients must be monitored by a medical 158 professional in order to begin an individualised nutritional approach.

is this part of any guidelines ? Seems a very definite statement when there is no individulatised nutritional approach

193

ONS does not 'complete the diet'. They help prevent dehydrations and replace electrolytes

Consider removing line 195-202 as not relevant - or shorten to one sentence.

Line 205 'These patients believe their GFDs positively influence their disease.....' This does not fit here as you further on in this paragraph quote study explaining this.

Line 210

suggest moving the hypothesis as to why gluten may help to the end of paragraph

237

There are conflicting results regarding the psychological state of IBD 237 patients on a GFD; some studies have shown that a GFD was responsible for a worse 238 overall psychological well-being, while in other studies adherence to a GFD has been 239 linked to lower rates of depression. In one study, elimination of gluten from the diet in 240 the paediatric population led to an improvement in depressive symptoms [51].

Needs better referencing

is this paediatric Crohn's population ?? the reference 51 is a review article and not the primary research

Line 261

Please add that ref 58 was a pilot study of patient with STABLE IBD.

Ref 57 is a poster reference

please change to published paper. Need to highlight again that this is in quiescent disease

Has been published Cox SR, Lindsay JO, Fromentin S, Stagg AJ, McCarthy NE, Galleron N, Ibraim SB, Roume H, Levenez F, Pons N, Maziers N, Lomer MC, Ehrlich SD, Irving PM, Whelan K. Effects of Low FODMAP Diet on Symptoms, Fecal Microbiome, and Markers of Inflammation in Patients With Quiescent Inflammatory Bowel Disease in a Randomized Trial. Gastroenterology. 2020 Jan;158(1):176-188.e7. doi: 10.1053/j.gastro.2019.09.024. Epub 2019 Oct 2. PMID: 31586453.

265

'This restricted diet has received considerable interest in its application during treatment. '

Remove this sentence

268

diary are not 'needed'  eg vegan diets - rephrase please

line 282

change 'including' to 'which includes'

Line 283

While dairy is a common 283 potential trigger food for IBD, - this is not correct . Please remove as is misleading. It suggests that diary causes a flare of IBD

Line 285 

A proportion of IBD patients believe that dairy products worsen symptoms and that 285 avoidance may help the disease. - reference

Line 323-327  repetition of what is in mediet - cut down

There is a lot sentences devoted to explaining the mediet , - considering reducing and keeping it more specific to IBD

Line 380- consider putting a paragraph break in this line

line 398- remove particularly in children as this is relevant to adults too

Line 424

There are 2 CD treat studies

You talk about the 5 cases in paeds but reference the adult paper which had more participants.

Please rectify- both should be referenced and discussed

Table 2- you state 'tomar et al' but then have 3 references . Suggest remove tomar et al. change to Up to 84%'

Meat alternatives - one study mentioned. Suggest including references from other studies and make a more general statement

Line 463- typo.

Second sentence in conclusion has already been stated in previous section.

line 469

I think you should differentiate diets for Treatment and diets for symptom management - two different things.

EEN /CD treat etc - for treatment

GFD , lactose free, low FODMAP etc- Main evidence is for symptom management -in patients with stable disease

line 484 - this is a summary paragraph . Consider revising this sentence

Line 49- no new ref in conclusions. Put ref 93 higher up eg section 3

table 1- fodmap can be used in patients with stable disease/in remission

Reviewer 2 Report

The manuscript needs some restructuring to improve readability. The sections are very long and dense, and should be broken up into individual paragraphs that concentrate on one point.

Section 3 on page 4 should include a discussion of eating disorders which complicate IBD.

Section 7 on page 11: The second sentence is uninterpretable.

Both tables are difficult to read because they include complete sentences and paragraphs. I suggest that division lines should be used to separate each section.

Specific recommendations (typos, recommended revisions):

Line 24: "affects" --> "affect"

Line 25: "rectolitis" --> "colitis"

Line 25: "The clinical manifestation" --> Each"

Line 27: "few are" --> "little is"

Line 32: "and considering the" --> "is considered to have"

Lines 33-34: "of this dietary pattern...of the diet itself" --> "but has low utilization in practice"

Line 35: "pointed on" --> "points out"

Line 47: "rectocolitis" --> "colitis"

Reviewer 3 Report

A very interesting and well written review article on the diet therapy of IBD patients. A very important aspect from a practical point of view is the issue of inflammatory bowel disease in pediatric patients, which occurs more and more often.

My suggestion is: (1) keywords “pediatric patient”, “FODMAP diet” should be added; (2) part 7.  Is rather “Summary and Conclusions”

Reviewer 4 Report

Only a few comments for this interesting review covering the wide topic of nutrition in IBD

  1. English need editing, line 47
  2. Can more be said about the pathogenesis in the introduction?
  3. When cyting references could you provide more detailed information in regards to exact magnitude of lowering the risk in percentage i.e.
  4. Table 1 and table 2 are not mentioned in the Manuscript body
  5. Line 105-111, please add references
  6. Line 146-150, it is somewhat unclear to what disease the mentioned drugs refer, please check and rephrase if necessary

Round 2

Reviewer 2 Report

Thank you for making the suggested revisions. The paper is much more readable.